# Few-Shot Learning for WiFi Fingerprinting Indoor Positioning

**DOI:** 10.3390/s23208458

**Published:** 2023-10-13

**Authors:** Zhenjie Ma, Ke Shi

**Affiliations:** School of Computer Science and Technology, Huazhong University of Science and Technology, Wuhan 430074, China; mazhenjie@hust.edu.cn

**Keywords:** indoor positioning, few-shot learning, prototypical networks

## Abstract

In recent years, deep-learning-based WiFi fingerprinting has been intensively studied as a promising technology for providing accurate indoor location services. However, it still demands a time-consuming and labor-intensive site survey and suffers from the fluctuation of wireless signals. To address these issues, we propose a prototypical network-based positioning system, which explores the power of few-shot learning to establish a robust RSSI-position matching model with limited labels. Our system uses a temporal convolutional network as the encoder to learn an embedding of the individual sample, as well as its quality. Each prototype is a weighted combination of the embedded support samples belonging to its position. Online positioning is performed for an embedded query sample by simply finding the nearest position prototype. To mitigate the space ambiguity caused by signal fluctuation, the Kalman Filter estimates the most likely current RSSI based on the historical measurements and current measurement in the online stage. The extensive experiments demonstrate that the proposed system performs better than the existing deep-learning-based models with fewer labeled samples.

## 1. Introduction

The increasing demand for indoor location-based services has driven a rapid development in indoor positioning techniques in recent years. Many practical applications, including emergency management [1], smart energy management, smart HVAC controls [2], point-of-interest identification, and occupancy prediction [3], have already emerged. A lot of indoor positioning solutions based on WiFi [4], Bluetooth [5], UWB [6], RFID [7], and Ultrasonic [8] have been proposed. WiFi is the most widely deployed indoor wireless infrastructure, and its signals are now ubiquitous in indoor environments. The WiFi positioning technique does not require additional hardware compared to the other solutions. It can provide indoor positioning by solely using the WiFi networks set up in indoor environments and the smart device at the user’s end. Although Bluetooth is popular in smart devices [9], Bluetooth-based methods require more base stations to cover the same area than WiFi-based methods due to the relatively short operating range.

WiFi-based positioning approaches can be divided into model-based and fingerprinting-based methods. Model-based methods utilize the propagation model of wireless signals in the forms of RSSI (Received Signal Strength Indicator), TOF (the time of flight), and/or AOA (angle of arrival) to obtain the distance and/or angle between the positioning device and WiFi AP (Access Point) and then use triangulation-based algorithms to decide the final position. Collecting AOA data requires building a matrix antenna and using the calculated angle to estimate the distance, which leads to high costs and complicated hardware [10]. Accurate TOF data collection highly depends on precise time synchronization [11]. In contrast, RSSI acquisition is simple and can be easily implemented on almost all devices. Most practical applications use RSSI as their data source.

Due to the wide fluctuation of WiFi signals [12] in indoor environments, the exact propagation model is challenging to obtain, which makes model-based approaches perform poorly in practical settings. To improve the positioning accuracy, fingerprinting-based methods [13] use the physically measurable properties of WiFi signals (usually RSSI) as fingerprints or signatures for each discrete spatial point to discriminate between positions. They adopt a two-stage mechanism: offline training and online positioning. In the offline phase, the RSSIs at the predefined RP (Reference Point) from several access points in the range are recorded and stored in a database along with their coordinates, which is called a site survey. This information can be deterministic or probabilistic. During the online positioning, the current RSSI vector at an unknown position is compared to those stored in the database, and the closest match is returned as the estimated user position.

In recent years, a variety of fingerprinting-based methods have been proposed. Deterministic methods [14,15] use Euclidean distance to express the difference between the measured RSSIs and stored RSSI fingerprints and apply the kNN (k-Nearest Neighbor)-based inferences to estimate the users’ positions. Probabilistic methods [16] collect RSSI distributions from the APs, store them as fingerprints, and then apply statistical inference such as Bayes rule and/or the maximum likelihood to estimate the users’ positions. Additionally, there are a lot of fingerprinting-based methods that use machine learning techniques, such as support vector machines [17], random forests [18], and artificial neural networks [19].

However, WiFi fingerprinting-based methods still suffer from vulnerable and changeable wireless signal transmissions. Some researchers have utilized CSI (Channel State Information) as the fingerprints since it contains richer information from multiple antennas and multiple subcarriers for accurate indoor positioning [20]. Unfortunately, CSI is only available with specific wireless network interface cards (NICs), e.g., Intel WiFi Link 5300 MIMO NIC (Intel, Santa Clara, CA, United States), and Atheros AR9390 or Atheros AR9580 chipset (Qualcomm, San Diego, CA, United States). RSSI still dominates in practical scenarios. Thus, in this paper, we adopt the RSSI for indoor positioning.

With the rapid development of artificial intelligence, deep learning has been successfully applied to numerous problems in many application areas. These include natural language processing, sentiment analysis, cybersecurity, business, virtual assistants, visual recognition, healthcare, robotics, and more. Due to their ability to approximate high-dimensional and highly nonlinear models, deep-learning-based models can improve positioning accuracy [21]. Recently, DNN (Deep Neural Network) [22,23], CNN (Convolutional Neural Network) [21], and LSTM (Long Short-Term Memory Network) [24] have been proposed to explore the underlying correlation between the RSSI data and the respective position to achieve a superior localization performance.

Deep-learning-based models require large amounts of labeled data to train. This kind of extensive data collection demands a time-consuming and labor-intensive site survey, which may be an expense not available for practical applications. The number of labeled samples is often limited, which may lead to overfitting of the trained model. Poor generalization has become a critical challenge that primarily affects the performances of deep-learning-based positioning models. Remarkably, the measured RSSIs in the online phase can have significant deviations from the RSSI fingerprints collected offline, seriously degrading the positioning accuracy.

In this paper, we leverage the wisdom of FSL (Few-Shot Learning) [25] to address the problems of positioning model robustness and insufficient sample size in WiFi fingerprinting-based indoor positioning. FSL applies meta-learning to perform learning tasks with few training examples. In the training process, meta-learning decomposes the dataset into different meta-tasks and learns the model’s generalization ability when the set of class labels changes. In other words, a machine learning model is trained using a series of training tasks. Here, each task mimics the few-shot scenario, which includes a certain number of classes with a few examples of each. These are known as the support set for the task and are used for learning how to solve this task. In addition, there are further examples of the same classes, known as a query set, which are used to evaluate the performance of this task. In the classical learning framework, we learn how to classify from training data and evaluate the results using test data. In the FSL framework, we learn how to classify given a set of training tasks (support sets) and evaluate using a set of test tasks (query sets). At each step of the training process, we update the model parameters based on a randomly selected training task. The loss function is determined by the classification performance on the query set of this training task based on the knowledge gained from its support set. Since the model is presented with a different task at each time step, it must learn how to discriminate data classes in general rather than a particular subset of classes. A PN (Prototypical Network) is a metric-based FSL method learning a well-generalized deep embedding network to map the examples into the embedding space. It assumes that each class can be represented by a prototype in the embedding space, takes the embeddings’ mean in this class as the prototype, and then uses the similarity between each prototype and the query’s embedding as a basis for classification.

We propose a novel WiFi fingerprinting-based indoor positioning system based on a PN, which learns a nonlinear mapping of the RSSI vector into an embedding space using a TCN (Temporal Convolutional Network) [26]. The position prototype is a weighted combination of its support set in the embedding space, where the weight is decided by the quality of the individual training sample characterized by its confidence region (represented by a covariance matrix). Online positioning is then performed for an embedded query point by simply finding the nearest position prototype. The intuition is to learn a position prototype for each position, which has smaller distances to training samples in the same position but larger distances to training samples in different positions. Since the training process is distance-based, it requires much fewer labeled samples than a traditional deep neural network.

Our positioning system keeps a short period of historical RSSI measurements. To mitigate the impact of WiFi signal fluctuation on the online measurement, KF (Kalman Filter) [26] estimates the most likely current RSSI based on those historical measurements and current measurements. Whenever the positioning object is static or mobile, KF can capture the temporal dependence and spatial correlation among the collected RSSI series to obtain a more accurate estimation and ease the mismatching problem.

We propose a PN-based positioning system that utilizes a distance-based loss function between the position prototype and the RSSI samples with the same position label to train a robust RSSI-position matching model with limited labels. To summarize, our work has the following main contributions:(1)We improve the traditional PN by using a TCN network as the encoder to learn an embedding. Compared to the commonly used CNN, the TCN-based encoder can utilize not only the signal strength, but also the relative strong or weak relationship among the strength of the WiFi signals received from different APs to construct a more robust mapping.(2)We improve the positioning prototype defining method. The extended PN can reflect the quality of individual samples by generating their embedding vectors and the confidence regions around them, characterized by a Gaussian covariance matrix. It allows a confidence-based weighting method in prototype defining, which can downweigh the contributions of low-quality samples through the covariance-based combination.(3)We use KF to estimate the most likely current RSSI based on the historical measurements and current measurements in the online stage to mitigate the adverse effects of the significant measuring errors.

The rest of the paper is organized as follows. Section 2 discusses related work, and our PN-based system is presented in Section 3. Section 4 gives the performance evaluation details. At last, Section 5 concludes the paper and points out the direction of future work.

## 2. Related Work

### 2.1. WiFi Fingerprinting

Recently, WiFi fingerprinting-based indoor positioning effectively gained accuracy improvement by utilizing deep learning approaches. A DNN was utilized to explore the underlying correlation between RSSI data and the respective position [22]. An algorithm reconstructing the missing data was also designed to achieve a superior positioning performance. A four-layer DNN structure [27], pre-trained by a stacked denoising autoencoder, was proposed to learn reliable features from a large set of noisy RSSI samples and avoids hand-engineering. WiDeep [24] combines a stacked denoising autoencoders deep learning model and a probabilistic framework to handle the noise in the received WiFi signal and capture the complex relationship between the WiFi AP signals heard by the mobile phone and its location.

CCpos [28] utilizes a convolutional denoising autoencoder to learn more robust invariant features and a convolutional neural network to achieve a high accuracy. In [24], a WiFi fingerprint-based localization algorithm using LSTM with explainable features and a sparse sample expansion algorithm based on a principal component analysis and Gaussian process regression for sparse samples was proposed.

Although the above methods utilize noise injecting [23,28], missing data reconstructing [22], and sparse sample enhancement [24] to preprocess the raw data samples and construct high-quality fingerprint databases and training data sets, these deep-learning-based models still suffer from data hunger caused by a high-cost site survey.

Mobility can increase localizability by indicating the physical relationships of the wireless signals between pairs of adjacent locations and extending the dimension of constraints for location estimation, which helps to distinguish multiple locations with similar RSSI fingerprints [29]. Soft-range limited kNN [30] exploits the information of previous positions and simultaneously applies the soft-range limiting factor for fingerprint distance calculation to achieve a more accurate and stable positioning performance. It requires that the speed of the targeting object is bounded.

Many researchers have realized that incorporating mobility information into a deep-learning-based positioning model can upgrade indoor positioning to a higher level. In [31], the proposed method used a sliding time window to build a temporal fingerprint chip as the input of the positioning model to extend the fingerprint diversity. The positioning model then used a TCN to extract spatiotemporal features from the finger chips and a DNN regressor to fit the complex nonlinear mapping relationship between the features and position coordinates. CTSLoc [32] uses a CNN model to extract the temporal fluctuation patterns of RSSIs and learn the nonlinear mappings from the signal features with time and space to position coordinates. The model input is a segment of RSSI trajectory data that is continuous in time, and the model output is the end position of the segment. Training a high-quality model requires large data samples, since different trajectories may end with the same end position.

TTSL [33] is a time-series-based localization framework that uses multiple consecutive RSSI measures to estimate the targets’ positions. It takes full advantage of the timing information attached to the RSSI vector in a trajectory. Utilizing a TCN to analyze the correlation among RSSIs in time and space attains a better performance. In [34], RNN (Recurrent Neural Network) solutions, including vanilla RNN, LSTM, GRU (Gated Recurrent Unit), bidirectional RNN, bidirectional LSTM, and bidirectional GRU, aimed at the trajectory positioning and took into account the correlation among the RSSI measurements in a trajectory. A weighted average filter was proposed for both the input RSSI data and sequential output locations to enhance the accuracy among the temporal fluctuations of the RSSI. These methods consider the information of the entire trajectory to improve the accuracy. However, they also demand many real trajectories representing the natural moving behaviors of positioning targets.

Different from the above methods, our proposed method utilizes the power of FSL to train a high-quality positioning model with limited labeled samples without the need to collect a large amount of RSSI data at each RP, label them with the positions, and/or assign them sequentially to the corresponding trajectory. KF is only applied in the online phase to filter the RSSI fluctuation and measuring noises, which is simple to implement and has a low resource consumption.

Some methods based on IMU (Inertial Measurement Unit) have been proposed for indoor positioning or tracking [35]. These methods are often called DR (Dead Reckoning) solutions. To achieve a better positioning accuracy, some researchers have combined WiFi fingerprinting and DR to determine the user position [36,37]. However, these methods rely on special hardware and inertial sensors, such as accelerometers, gyroscopes, and magnetometers, to track the targets. Besides RSSI data, IMU data must also be collected, leading to more resource consumption.

### 2.2. Few-Shot Learning

Deep learning is a data-hungry method which requires a large amount of data to decide the optimized model parameters to achieve a good performance. This parametric deep learning could perform better in limited labeled data settings. Inspired by humans’ robust reasoning and analytical capabilities, the concept of FSL [25] has emerged and provides a promising way to handle data scarcity scenarios. A PN [38] is a metric-based few-shot learning method that projects samples into a metric space learned by a neural network, making similar samples closer and various samples farther apart. In this way, the classification problem is turned into the nearest neighbor problem in the metric space.

Recently, some PN-based approaches have been proposed to solve the classifying of problems in real-world applications. TapNet [39] is a novel multivariate time series classification model based on an attentional prototype network capable of training the feature representation based on their distance to class prototypes with inadequate data labels. In [40], a PN-based model was proposed to realize high-accuracy intelligent fault diagnosis based on small samples and intense noise problems. The model presented in [41] introduced a hybrid attention module and combined it with prototypical networks for few-shot sound classification. This hybrid attention module consisted of feature- and instance-level attention blocks. This two-attention mechanism can highlight key embedded features and emphasize crucial support instances.

Some researchers have utilized few-shot learning to solve indoor positioning problems. In [42], the authors introduced the concept of few-shot learning into transfer learning for device-free indoor localization using CSI. It can significantly reduce the data collection and labeling costs for localization by using only a small amount of labeled data from the current environment and reusing a large amount of existing labeled data previously collected in other environments. Therefore, the main focus is domain transferring. STONE [43] is a framework that delivers stable and long-term indoor localization without any retraining. It adopts a Siamese neural encoder-based network to deal with the degradation of localization accuracy caused by AP replacement or removal over time. Our method uses prototypical networks that do not require obtaining pairs of examples and evaluating individual pairs selected from all classes during training.

## 3. Prototypical Networks Based Few-Shot Learning for Indoor Positioning

In this section, we elaborate on the proposed scheme, whose core method lies in PN-based few-shot learning. To enhance the accuracy among the RSSI fluctuations, we adopt KF to estimate the most likely current RSSI based on the historical measurements and current measurements in the online stage.

### 3.1. Overview

The general framework of our proposed scheme is a PN-based few-shot learning model. The schematic of the proposed indoor positioning is shown in Figure 1. WiFi fingerprinting-based indoor positioning is a typical classification problem. The input is RSSI vector $X=\left\{x_{1},\ldots ,x_{l}\right\}$, where *l* represents the number of APs and *x_i_* represents the RSSI value from the *i*th AP. The output is a specific position label defined by $Y=\left\{y_{1},\ldots ,y_{n}\right\}$, where *n* represents the number of RPs and *y_i_* indicates whether it is located at the *i*th RP. Positioning is learning a target function to map each X to one of the predefined position labels Y. To reduce the impact of overfitting caused by limited labeled data, the proposed method adopts metric-based FSL that utilizes a TCN to project samples into a metric space, which makes similar samples closer and a variety of samples farther apart in the space. In this way, the positioning problem is turned into the nearest neighbor problem in the embedding space. The positioning procedure includes offline model training and online position estimation.

In the offline phase, we use a TCN as an embedding function $f_{\theta }:\;R^{F}\to R^{D}$ with learnable parameters θ to project the samples in the $R^{F}$ into the high-dimensional embedding vectors (dots in the embedding space in Figure 1) in the embedding space $R^{D}$. The training aims to find an optimized θ to make the samples from the same position closer and the samples from the different positions farther away from each other in the embedding space. Instead of using a fully connected layer and SoftMax loss function to decide the position, the proposed method utilizes distance-based similarity measurement.

We assume that we have a training dataset $C_{batch}=\left\{c_{1},c_{2},c_{3},\ldots ,c_{n}\right\}$ in the training process, where *n* is the number of sample classes (it is the number of RPs in the positioning). Since FSL trains the model with a series of training tasks, random selection generates the support sets and query sets corresponding to these tasks. In the random selection process, a portion of each class of samples in $C_{batch}$ will be randomly selected to form the support set $S=\left\{s_{1},s_{2},s_{3},\ldots ,s_{m}\right\}$, and then the remaining samples will be formed into the query set $Q=\left\{q_{1},q_{2},q_{3},\ldots ,q_{k}\right\}$. For each sample in the support set, an embedding vector (dot in the embedding space in Figure 1) and its corresponding covariance matrix indicating the confidence region (ellipse around the dot in the embedding space in Figure 1) are computed by the embedding function $f_{\theta }$. Each position’s prototype (star in the embedding space in Figure 1) is a variance-weighted linear combination of the embedding vectors of individual support samples. Support sets are used to define the prototypes and covariance matrices of the particular position. Different colors represent different classes. We use a query set to train the model. The samples in the query set are also projected into the embedding space, and the distances between the query embeddings and position prototypes are used for loss calculation and classification. The training process aims at determining the optimized parameter to minimize the loss.

In the online phase, we collect a series of RSSI measurements $X^{T}=\left\{X_{1},\ldots ,X_{T}\right\}$ using continuous sampling in a time window, where *T* is the length of the time window. Then, the KF module uses this series of measurements, containing noise and other inaccuracies caused by wireless signal fluctuations, to produce the most likely RSSI at the current position. Finally, the online position-matching model trained in the offline phase takes this estimated RSSI vector as the input and generates the position.

### 3.2. TCN-Based Encoder

We use a multi-layer TCN without an explicit fully connected layer to encode the RSSI vectors into high-dimensional Euclidean vectors. The encoder can be defined as $encoder\left(\theta \right):\;I\in R^{F}\to \vec{x}\in R^{D}$.

In WiFi fingerprinting, an RSSI vector collected at an RP contains a series of RSSI values measured from different APs. There are spatial relations among these RSSI values. For example, an RSSI value measured from an AP closer to the RP will be larger than an RSSI value measured from an AP farther away from the RP. Compared to the RSSI values, the relatively strong/weak or rank information decided by the spatial distribution is less impacted by environment dynamics and signal fluctuations. It tends to be more inconsistent.

We use a TCN, an effective and powerful sequence modeling model, as the prototypical network’s encoder to extract the spatial information. In contrast to the frequently used CNN model in the positioning, the distinguishing characteristics of the TCN, such as dilated and causal convolutions, make it more appropriate to extract the series information. While the TCN holds the same flexibility in taking a sequence of any length and giving rise to an output sequence of the same length and has the same capacity to extract the series pattern information as the RNNs, it adopts a simple architecture. It consumes fewer resources to train the model.

As shown in Figure 1 and Figure 2, the RSSI vector $X=\left\{x_{1},\ldots ,x_{l}\right\}$ is the input of the based encoder, which consists of a series of residual blocks with identical structures. The residual connection mechanism prevents network degradation, effectively solving gradient disappearance and explosion problems. Each residual block contains two layers, namely causal and dilated convolutions. The causal convolution mechanism adds causal constraints, and the connections that do not meet the dependence constraint are removed in the training and learning processes. Through the dilated convolution mechanism, larger receptive fields of information are obtained without increasing the number of network layers. Here, *k* represents the size of the convolution kernel used for information extraction and *d* denotes the dilation coefficient. The size of the convolution kernel of these residual blocks is identical. The dilation coefficient of these residual blocks increases exponentially, which causes the information receptive field to expand twice, in turn.

A weight-normalized layer is added after each causal and dilated convolution layer to speed up the model convergence. The ReLU activation layer activates neurons nonlinearly and adds nonlinear factors to the model to improve its feature learning ability. The Dropout layer sets the loss rate to inactivate neurons proportionally and randomly to avoid an excessive fitting model.

### 3.3. Prototype Defining

Prototype defining can be viewed as clustering on the support set, with one cluster per class and each support point assigned to its corresponding class cluster. In the typical prototypical networks, each prototype is the mean vector of the embedded support points belonging to its class. It assumes that each sample in the support set has the same contribution to the final position prototype. However, this assumption may not hold for indoor positioning due to the complex environment and device dynamics. At the same RP, the collected samples have different errors. Samples with smaller errors should have more contribution to the final position prototype, and samples with larger errors should have less contribution to the final position prototype. Therefore, our PN-based model extends the typical prototypical network architecture to allow the model to reflect the quality of individual samples by predicting their embedding vectors and the confidence regions around them.

We define the extended model as:(1)$$encoder\left(\theta \right):\;I\in R^{F}\to \left[\vec{x},{\vec{s}}_{raw}\right]\in \left[R^{D},R^{1}\right]$$
where $\vec{x}\;$is the generated embedding vector, ${\vec{s}}_{raw}$ characterizes the predicted size of the confidence interval around this generated embedding vector, and $R^{1}\;$is the dimensionality of the predicted components of the covariance matrix. The covariance matrix is defined as:(2)$$\sum =diag\left(\sigma ,\sigma ,\sigma ,\sigma ,\ldots \right)$$
where σ is calculated from the raw encoder output ${\vec{s}}_{raw}$. Then, we translate the raw covariance matrix output of the encoder into an actual covariance matrix by using the SoftPlus function defined as:(3)$$S=1+softplus\left(S_{raw}\right)=1+log\left(1+e^{S_{raw}}\right)$$

It is applied component-wise. Since $softplus\left(x\right)$ > 0, this guarantees *S* > 1, and the encoder can make data samples less important.

In our PN-based model, a critical part is the creation of a position prototype from the available RSSI data samples collected at an RP. We propose a variance-weighted linear combination of embedding vectors of individual data samples as our solution. Let Position *C* have RSSI vectors $X_{i}$ that are encoded into embedding vectors ${\vec{x}}_{i}^{c}$, and inverses of covariance matrices $S_{i}^{c}$, whose diagonals are ${\vec{s}}_{i}^{c}$. The position prototype, i.e., the centroid of the class, is defined as:(4)$${\vec{p}}_{c}=\frac{\sum_{i}{\vec{s}}_{i}^{c}\circ {\vec{x}}_{i}^{c}}{\sum_{i}{\vec{s}}_{i}^{c}}$$
where ◦ denotes a component-wise multiplication, and the division is also component-wise. The diagonal of the inverse of the class covariance matrix is then calculated as:(5)$${\vec{s}}_{c}=\sum_{i}{\vec{s}}_{i}^{c}$$
which corresponds to the optimal combination of Gaussians centered on the individual data samples into an overall class Gaussian. This allows the model to down-weight less important examples in defining the class. Therefore, the data samples with more significant errors would have less contribution to the final position prototype determination, which makes our model more suitable for indoor positioning environments.

As shown in Figure 3, an RSSI vector is mapped to its embedding vector (dot) by the encoder. Its covariance matrix (ellipse) is also output by the encoder. An overall covariance matrix for each class is then computed (large ellipses), as well as prototypes of the classes (stars). Different colors represent different classes. 

### 3.4. Similarity Measurement and Training

We utilize linear Euclidean distances to measure the similarity between the data samples and the position prototypes. The encoded embeddings of the data examples are used to define where a particular RP *C* lies in the embedding space. The distances between the query examples and the RPs’ prototype determined from the support points are then used to classify and calculate a loss. The distance $d_{c}\left(i\right)$ from a position prototype *C* to a query point *i* is calculated as:(6)$$d_{c}\left(i\right)=\sqrt{{\left(\vec{x}-{\vec{p}}_{c}\right)}^{T}S_{C}\left(\vec{x}-{\vec{p}}_{c}\right)}$$
where ${\vec{p}}_{c}$ is the centroid, or prototype, of position *C*, and $S_{C}\;$is the inverse of its covariance matrix. Therefore, the model can learn class-dependent distance metrics in the embedding space.

As shown in Equation (7), we predict a query sample’s position as a position with the minimum distance from the position prototype.
(7)$$\hat{y}\left(i\right)={argmin}_{c}d_{c}\left(i\right)\;$$

The speed of the training and its accuracy depend strongly on how distances are used to construct a loss. The specific form of the loss function is the combination of SoftMax with cross-entropy, as shown in Equation (8).
(8)$$Loss=\frac{1}{N_{Q}}\sum_{i}\left(-log\left(\frac{exp(-d_{y}\left(i\right))}{\sum_{k}exp(-d_{k}\left(i\right))}\right)\right)$$

In the training process, the embedding vectors and their confidence estimations of the data samples are continuously updated with the model parameters’ change by minimizing the loss function. Then, the prototype center is updated so that the confidence of the position prototype is improved. In contrast to typical prototype networks, the encoded output of the embedding function is interpreted as the embedding vector of the sample and its confidence estimation, where the confidence is represented using a radius component of the covariance matrix. Therefore, iterative training can down-weight the low-quality data samples, which makes the final prototype more reasonable.

### 3.5. Online RSSI Data Filter

In the online phase, the PN-based model trained in the offline phase receives the RSSI data sample as its input, calculates the distance between the position prototypes of RPs and these RSSI embeddings, and predicts the position of this RSSI embedding as an RP that has a minimum distance.

However, RSSI values are heavily influenced by the environment and have high noise levels. For example, multi-path reflections may make signals bounce against objects in the environment, such as walls and furniture, which leads to a more significant deviation. Although the PN-based model proves to be more robust, the large input error may cause significant positioning errors, leading to mismatching problems.

To address this issue, our positioning system keeps a short period of historical RSSI measurements. It utilizes KF to estimate the most likely current RSSI based on historical and current measurements. KF is a state estimator that estimates unobserved variables based on noisy measurements. It is a recursive algorithm as it considers the history of measurements. In our paper, we collect a series of RSSI measurements $X^{T}=\left\{X_{1},\ldots ,X_{T}\right\}$ using continuous sampling in a time window, where *T* is the time window length. We want to obtain the true RSSI based on this series.

The use of the KF algorithm includes two stages: prediction and correction. In the prediction phase, the filter takes the estimate of the previous state as its input and outputs the predicted value of the current state. In the correction stage, the expected value obtained in the prediction stage is corrected according to the observed value of the current state. After several iterations, an optimal estimated value that is infinitely close to the true value is obtained.

## 4. Performance Evaluation

In this section, we verify the effectiveness of the proposed method based on the data obtained from the actual environments and make comparisons with other existing methods. We also analyze the factors that influence the performance of our system.

### 4.1. Setup

The experimental environment is illustrated in Figure 4. The experiment area covers approximately 164 square meters, including rooms A, B, C, and E, and corridor B. There are eight APs (indicated by the red stars in Figure 4) deployed in the experiment area. The grid of RPs in the environment area includes 111 points with spacings of 1.5 m (indicated by the blue spots in Figure 4). During the offline phase, we collected 110 RSSI samples at each RP. During the online phase, we set the time window to 10, meaning each positioning target keeps 9 historical RSSI measurements and 1 current RSSI measurement. The sampling interval is set to 1 s, which allows us to obtain the latest RSSI value from the underlying hardware. Otherwise, we may obtain the cached RSSI values from the previous measurement instead of the current one.

The metric most interesting for all positioning systems is the position error. Considering a two-dimensional environment area, the position error is defined as the Euclidean distance between the actual physical and estimated positions. The average positioning error is calculated as:(9)$$error=\frac{1}{n}\sum_{i=1}^{n}\left\|\hat{y_{i}}-y_{i}\right\|$$
where *n* is the number of query points, $\hat{y_{i}}$ is the estimated position of the *i*th query point, and $y_{i}$ is the actual position of the *i*th query point. We also use the error’s CDF (Cumulative Distribution Function) for a more comprehensive analysis.

Our PN-based model uses the TCN network as the encoder, which contains five residual blocks, and the specific parameters of each residual block are shown in Table 1. Each residual block has two identical convolutional layers, and each convolutional layer is followed by an activation layer and a Dropout operation to avoid model overfitting. The convolution kernel size of each residual block is 5, and the activation function is ReLU. The convolution kernels of the first two residual blocks are 32, and the probability of random inactivation (Dropout) is 0.2. The parameters of the last three residual blocks are the same. The probability of random inactivation is 0.5, and the number of convolution kernels is 64. The only exception is the number of convolution kernels of residual block 5. This number is 65 because the output embedding vector combines 64-dimensional embedding feature vectors and 1-dimensional vectors used to calculate the covariance matrix. The dilation coefficients of the convolutional layers in each residual block are 1, 2, 4, 8, and 16, respectively.

We implement the proposed model in PyTorch. We allocate 70% of the available data for training, and the remaining 30% are test data sets. We use Adam (Adaptive Moment Estimation) as our optimizer when training the model.

### 4.2. Performance Comparison

Deep learning methods have been widely used in WiFi fingerprinting-based indoor positioning and show a superior performance compared to non-deep learning methods. Hence, deep learning methods are chosen as comparators. To avoid collecting extra trajectory data, our PN-based method focuses on RSSI-position matching. Therefore, deep learning models incorporating mobility information are not considered. Under this circumstance, DNNs and CNNs are the most used methods and provide the current state-of-the-art performance. Although the methods proposed in [42,43] use few-shot learning strategies, they mainly focus on cross-domain adaption with transfer learning or contrastive learning, which are beyond the scope of our paper. We will address this issue in future work. Finally, we compare the performance of our PN-based method with a DNN [19] (adopting stacked denoising autoencoders) and CNN [24] (adopting ResNet architecture)-based methods.

Figure 5 illustrates the comparison results, where *m_p_* is the number of support samples for each position. We can see that our method performs better than these methods. The CDF curve of the PN is steeper than that of the DNN and CNN, regardless of the value of *m_p_*. Compared to the DNN and CNN, the performance gain of the PN increases as the number of support samples decreases. When the value of *m_p_* is 10, the average errors of the DNN and CNN are 4.2 m and 4.4 m, respectively, twice as much as the average error of the PN, 2.3 m. When the value of *m_p_* is 25, the average errors of the DNN and CNN are 3.1 m and 3.3 m, respectively, two times larger than the average error of the PN, 1.19 m. This proves that a small number of samples can cause the trained DNN and CNN model to overfit, which leads to a larger error. The PN-based model can take the advantages of few-shot learning to achieve a low error with a few samples. As the value of *m_p_* increases, the errors of the DNN and CNN are reduced. When the value of *m_p_* is 50, the average errors of the DNN and CNN are 2.69 m and 2.3 m, respectively. When the value of *m_p_* is 100, the average errors of the DNN and CNN are 2.57 m and 1.17 m, respectively. The CNN performs better than the DNN due to its residual architecture allowing a better application of the backpropagation training. Our PN still performs the best. When the values of *m_p_* are 50 and 100, the average errors of the PN are 0.98 m and 0.96 m, respectively.

The above results and analysis demonstrate that projecting the RSSI vectors into the embedding space makes their difference more noticeable, which mitigates the position ambiguous problem and reduces positioning errors. Using distance-based metrics to measure similarity can train a high-quality positioning model with few samples.

In the following subsections, we will investigate the impacts of the encoder selection, prototype-defining methods, and online filtering methods on the performance.

### 4.3. Impacts of Encoder Selection

In our prototype network, the encoder projects the original RSSI vectors into the embedding space so that a distance-based classifier can recognize them. The embedding vectors mapped from the RSSI vectors collected in the same position should be close to each other in the embedding space, and the embedding vectors mapped from the RSSI vectors collected in the different positions should be far away from each other in the embedding space. The encoder should be capable of learning stable and discriminative features.

To investigate the impacts of the encoder selection on the final positioning error, we compare our TCN-based encoder with CNN-based and LSTM-based encoders. Figure 6 illustrates the comparison results. A CNN is a widely used encoder structure in prototypical networks. Here, the compared CNN has eight convolutional layers, and the convolutional kernels in these layers are 16, 16. 32, 32, 32, 64, 64, and 65, respectively. The size of the convolutional kernel is 5, and the size of the pooling kernel is 3. LSTM is widely used in sequence processing tasks. The compared LSTM has 100 hidden units.

In indoor positioning, a CNN can capture the relationships between the signal strength and physical distance as discriminative features to recognize different positions. LSTM can capture ranking information and the relative strong or weak relationships among the signal strengths from different APs to identify different positions. Compared to ranking information, the strength–distance relationship is more distinctive in position differentiation and less likely to cause position ambiguity, which is why the average error of the CNN-based encoder is lower than that of the LSTM-based encoder. However, the signal strength is vulnerable to environmental changes and interference, while ranking information is more stable. When the number of support samples is small, the signal strengths become less reliable, degrading the CNN-based encoder’s performance. At the same time, relatively stable ranking information makes the LSTM-based encoder perform similarly. Therefore, the average errors of these two encoders are very close. Our TCN-based encoder has the lowest error among these three encoders due to the dilation and casual convolution, which allow it to capture the strength and ranking information simultaneously to differentiate the positions. When the number of support samples is below or equal to 50, the TCN-based encoder has an approximately 20% improvement in its positioning accuracy compared to the CNN-based encoder. When the number of support samples exceeds 50, the positioning accuracy still has approximately a 10% improvement.

### 4.4. Impacts of Prototype Defining

In the typical prototypical networks, each prototype is the mean vector of the embedded support points belonging to its class. This defining method makes each sample in the support set contribute equally to the final position prototype, regardless of its quality. The existence of low-quality samples will lead to a significant deviation between the computed prototype and the actual cluster center, degrading the accuracy of distance and adversely affecting the performance. Our confidence-based weighting method can downweigh the contributions of low-quality samples through the covariance-based combination.

As shown in Figure 7, the average error of the confidence-based method is lower than that of the mean vector method, no matter the number of support samples. When the number of support samples is small, a few low-quality samples may have significantly negative effects on the results. Therefore, the error decreases more significantly. When the number of support samples is 10, the average error decreases from 2.9 m to 2.37 m. When the number of support samples is 25, the average error decreases from 1.7 m to 1.17 m. When the number of support samples is 50, the average error decreases from 1.3 m to 0.98 m. Although the decreasing trend slows with increasing support samples, the confidence-based method performs better than the mean vector method. When the number of support samples is 100, the average error decreases from 1.09 m to 0.96 m. This is because the negative effect of low-quality samples still exists, and our method can mitigate this effect.

### 4.5. Impacts of Online Filter

Figure 8 compares the average errors among the proposed PN, CNN, and LSTM models with and without KF filters in the online positioning phases.

In these three models, the KF filter decreases the average errors no matter the number of support samples. Our PN-based method projects the original RSSI vectors into ahigh-dimensional embedding space. It utilizes distance-based metrics to recognize the positions to improve the model’s generalization ability, which can reduce the impacts of environmental noise and interferences on the accuracy. However, in WiFi fingerprinting-based indoor positioning, many uncertain factors, such as signal transmitting jitter and instantaneous environmental changes, can significantly fluctuate a single measurement. Sometimes, this fluctuation may cause the measured value to deviate from the actual value of the target position but close to the actual value of another position, which decreases the positioning accuracy. KF is a powerful tool for estimating and predicting system states in uncertainty, which can mitigate this negative impact. In CNN and LSTM models, KF can decrease the average errors for the same reason.

## 5. Conclusions and Future Works

This paper presented a prototypical-network-based positioning system, which explored the power of few-shot learning to establish a robust RSSI-position matching model with limited labels. A TCN encoder learnt an embedding of individual RSSI vectors collected at the RP and its quality. On the embedding space, each RP had a corresponding prototype defined as a weighted combination of the embedded RSSI vectors belonging to this RP. Online positioning was performed for embedded query RSSI vectors by simply finding the nearest position prototype. To mitigate the space ambiguity caused by signal fluctuation, the system used Kalman Filter to estimate the most likely current RSSI based on the historical measurements and current measurements in the online stage.

Our PN-based method focused on constructing a robust and generalized RSSI-position matching model with limited labels. It could handle short-term signal fluctuation very well using a TCN-based encoder, confidence-based prototype defining, and a KF-based online data filter. However, our PN-based model may degrade significantly when indoor environments undergo significant long-term changes such as large changes in furniture, equipment arrangement, and AP removal or replacement. We need to collect new data and retrain the model to maintain its accuracy in this changed environment. To address this issue, we will investigate incorporating a transfer-learning-based domain adaption method or a contrastive-learning-based unsupervised method in future work. In addition, KF works well only when the target is static or moves smoothly without sudden changes. When the target’s movement does not fall into this pattern, we may need to design a more supplicated online data-filtering method to reduce the input error in the online stage. If we can obtain the trajectory data, incorporating mobility information may improve the accuracy of our PN-based method further.

## Figures and Tables

**Figure 1 sensors-23-08458-f001:**
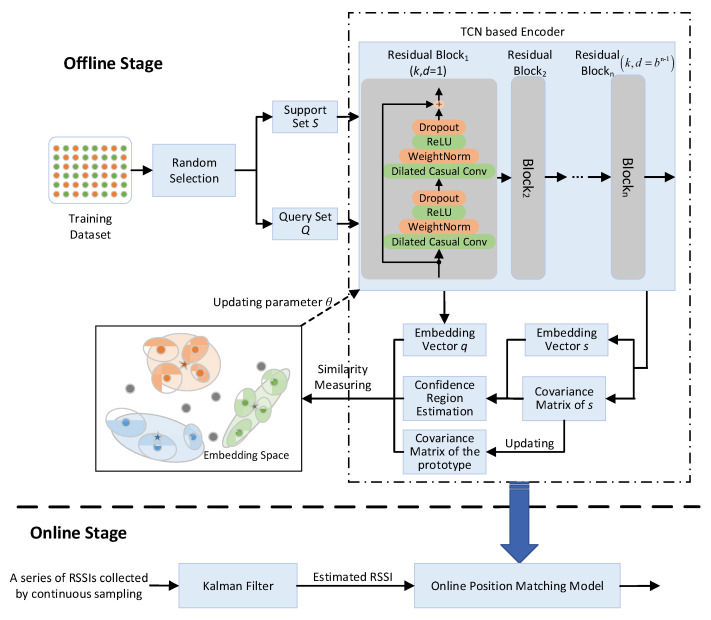
The overall architecture of the PN-based indoor positioning.

**Figure 2 sensors-23-08458-f002:**
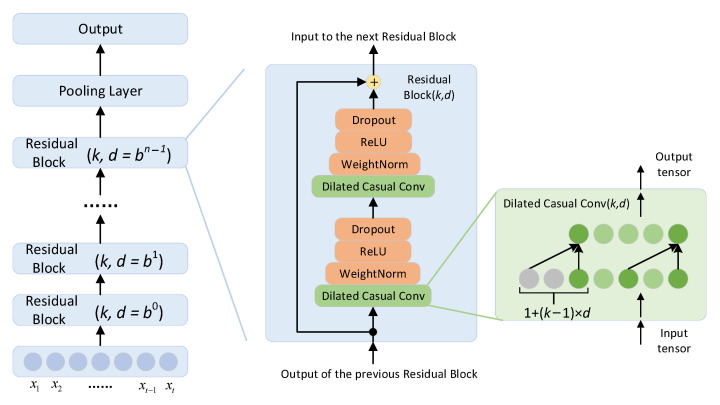
TCN encoder architecture.

**Figure 3 sensors-23-08458-f003:**
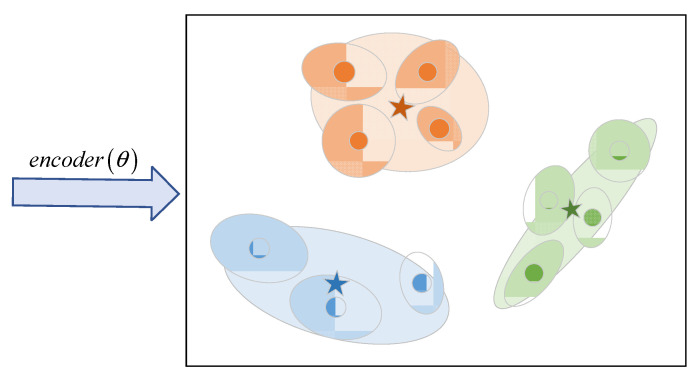
A diagram showing the embedding space of our model.

**Figure 4 sensors-23-08458-f004:**
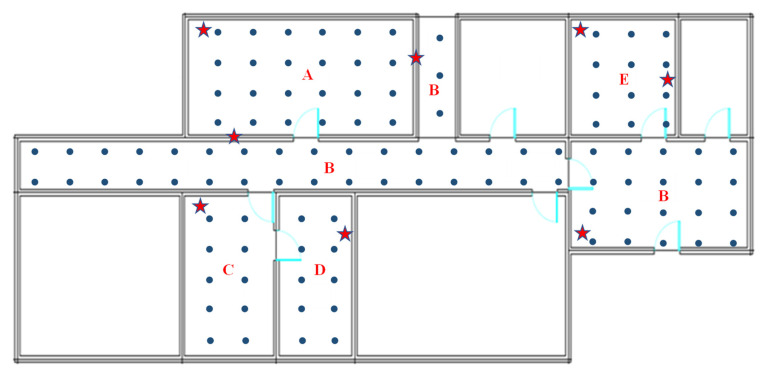
Experimental environment.

**Figure 5 sensors-23-08458-f005:**
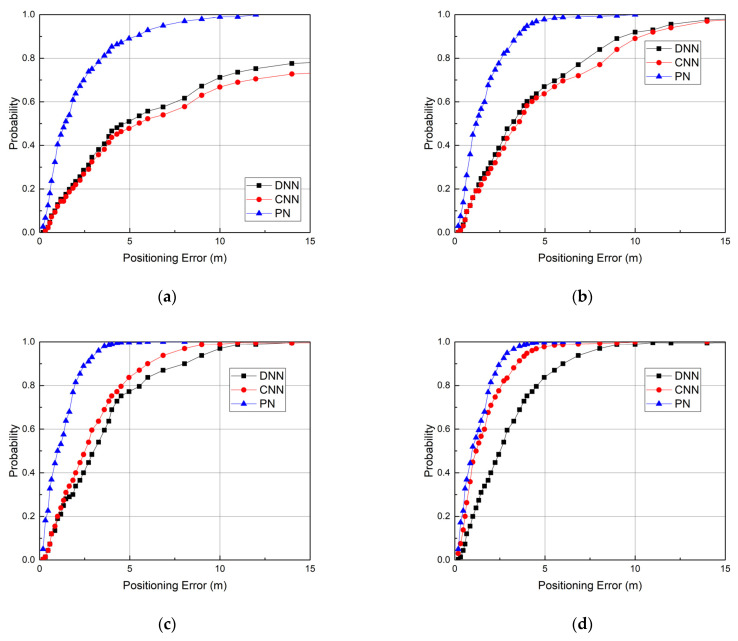
Performance comparison. (**a**) *m_p_* = 10, (**b**) *m_p_* = 25, (**c**) *m_p_* = 50, and (**d**) *m_p_* = 100.

**Figure 6 sensors-23-08458-f006:**
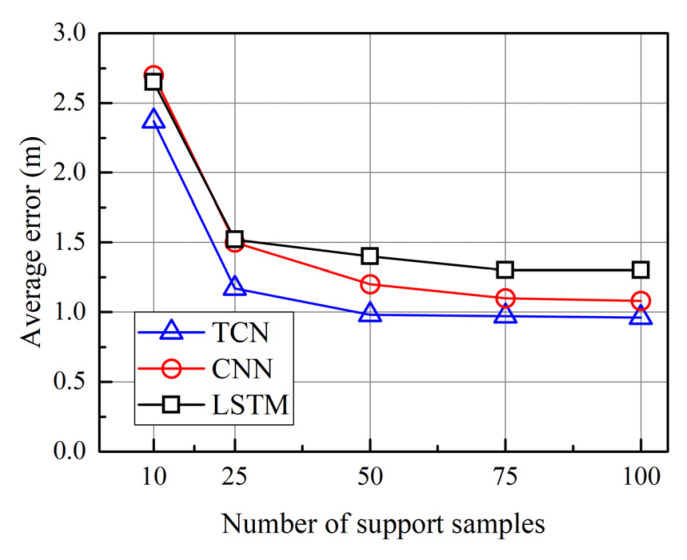
Impacts of encoder selection.

**Figure 7 sensors-23-08458-f007:**
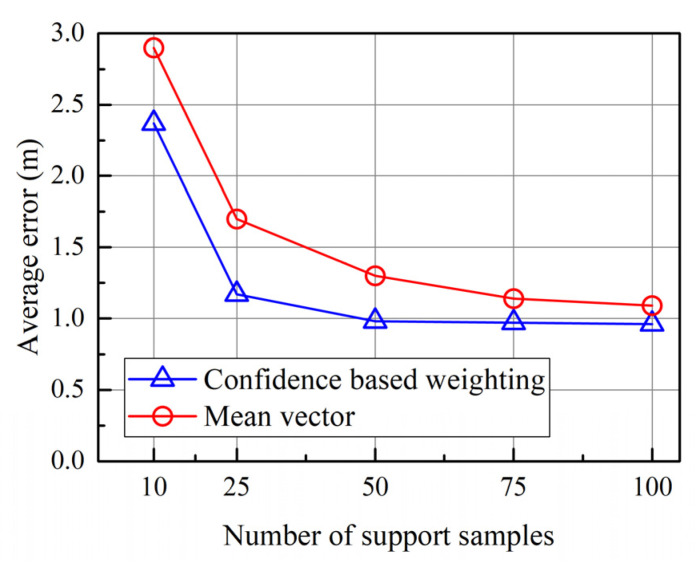
Impacts of prototype defining.

**Figure 8 sensors-23-08458-f008:**
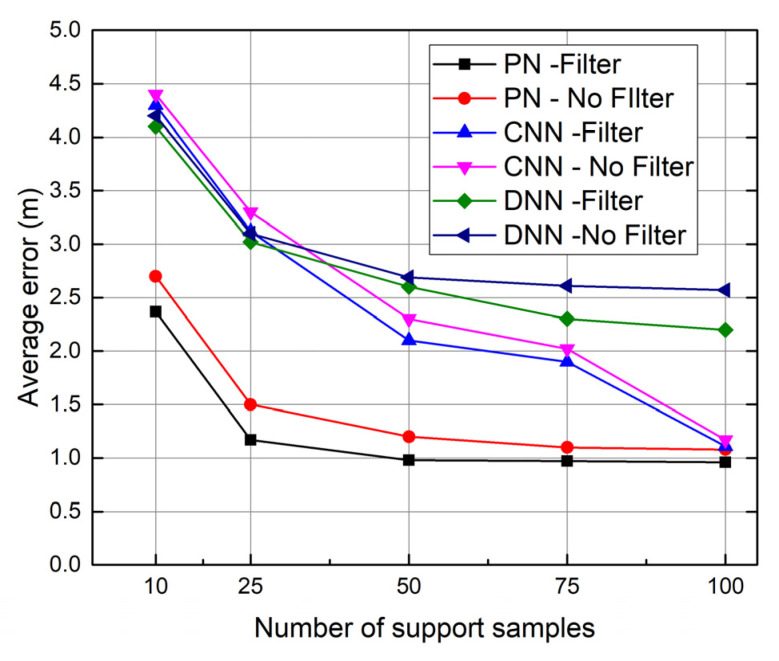
Impacts of online RSSI data filter.

**Table 1 sensors-23-08458-t001:** Summary of TCN structure and parameters.

	Convolutional Layer Kernel Size Kernel Number		Activation Function	Dropout	Dilation Coefficient
Residual Block 1	5	32	ReLU	0.2	1
Residual Block 2	5	32	ReLU	0.2	2
Residual Block 3	5	64	ReLU	0.5	4
Residual Block 4	5	64	ReLU	0.5	8
Residual Block 5	5	65	ReLU	0.5	16

## Data Availability

The data used to support the findings of this study are available from the corresponding author upon request.

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
