# Peer review of "Few-Shot Learning for WiFi Fingerprinting Indoor Positioning"

_sensors, 2023, doi:10.3390/s23208458_

Round 1

Reviewer 1 Report

Comments to the Author

This paper proposes a prototypical network-based positioning system, which explores the power of few-shot learning to establish a robust RSSI-position matching model with limited labels. It is an interesting research topic with many potential application areas. However, there are several points that need to be addressed to improve the quality of the manuscript.

Suggestions to improve the quality of the paper are provided below:

1.     It was mentioned very briefly in the first paragraph of the Introduction section that there is an increasing demand for indoor location-based services that has driven a rapid development in indoor positioning techniques. However, the authors did not go into any details about the applications of indoor localisation. Some of the popular applications that come to mind include emergency management, smart energy management, smart HVAC controls, point-of-interest identification, and occupancy prediction. I suggest that the authors review the following established works as a good starting point and highlight some of these application areas where indoor positioning systems are leveraged.

Indoor localisation for building emergency management

10.1109/IUCC-CSS.2016.013

Indoor localisation for smart energy management

Indoor localisation for smart HVAC controls

https://doi.org/10.1145/2517351.2517370

Indoor localisation for point-of-interest identification

Indoor localisation for occupancy prediction

https://doi.org/10.1016/j.buildenv.2022.109689

2.     While the authors have listed out a few indoor localisation technologies in the Introduction section (e.g., Bluetooth, UWB, RFID, and Ultrasonic), the current content is very lacking as the authors directly go on to highlight the advantages of Wifi-based approaches, without any discussions about the other indoor localisation technologies. Out of this list of technologies, Bluetooth Low Energy-based approaches are identified by [1] to be particularly popular due to their ubiquitous nature and does not require additional hardware, similar to Wifi-based approaches. Please kindly review the following paper as a good starting point and compare between these two technologies (i.e., Wifi and Bluetooth Low Energy), before concluding why Wifi is selected for this study.

[1] https://doi.org/10.1016/j.buildenv.2020.106681

3.     Please rephrase the contributions paragraph in the Introduction section to more clearly highlight the ways in which the proposed approach improves upon the previous works, instead of what was done in this paper.

4.     The literature review for Section 2.2 seem to be a general introduction of few shot learning and different approaches for performing few shot learning. However, the current literature review does not mention anything about past indoor localisation studies that have also adopted few shot learning. Please review [1] and [2], and discuss about how the proposed approach extends upon these works.

[1] https://doi.org/10.1109/ICC45855.2022.9839217

[2] https://doi.org/10.1007/978-3-031-26712-3_17

5.     It is unclear why the authors decided to compare the performance of the proposed approach against CNN and DNN. Are these two methods considered as the current state-of-art? What about the models proposed on in the studies listed in Comment 4?

6.     Please include a Discussion section before the Conclusion section, then provide a detailed discussion about the limitations of this work and how they will be addressed in future works.

7.     Minor comments

·       The legend label for Figure 5a is wrongly labelled. Should be “DNN” instead of “CNN” for the black trend line.

·       For Figure 5, include the measurement unit for Positioning Error in the figure.

There are no major issues related to the manuscript's quality of English, except for some minor issues highlighted in my current set of comments.

Reviewer 2 Report

The article addresses the issues of high indoor fingerprint positioning sample acquisition costs and poor robustness due to wireless signal fluctuations. It proposes a location-based method based on prototype networks. This method applies the concept of few-shot learning, establishing a location matching model based on limited labeled RSSI data. It utilizes a TCN encoder to map individual RSSI samples into a feature space and combines features belonging to the same location with weights to obtain prototype features for that location. At Online phase, a Kalman filter is used to estimate the optimal RSSI for the test location based on continuously collected data, achieving positioning through matching the prototype features with various AP points. This method has achieved good positioning results with limited labeled samples, improving positioning accuracy. The article is innovative and well-substantiated in its experimental content, but it lacks clarity in describing the model design and contains some writing errors, as detailed below:

a. The authors state in the abstract that the article innovatively applies the concept of few-shot learning to construct a prototype feature extraction model. However, they did not introduce or explain the principles of few-shot learning, such as the relationship between support sets and query sets or other necessary prior knowledge. This omission leads to a lack of coherence and completeness in the article. It is recommended that the authors add a section in the first chapter to provide an introduction to few-shot learning, addressing these aspects for a better understanding of the readers.

b. In section 3.1 of the article, on line 232, it is mentioned that the method randomly samples support set S and query set Q from the training set Cbatch. However, this explanation does not clarify the relationship between the query set, support set, and training set, making it difficult for readers to understand the feasibility and effectiveness of designing a prototype feature extraction model based on the concept of few-shot learning. It is recommended to provide additional clarification in this section to better convey the design rationale of the model and enhance the readability of the article.

c. In the article, in the third section, Figure 1 shows a connection from the "Embedding Space" module leading to the "Updating parameter θ" module, while this update should be related to the parameters in the feature extraction network "TCN-based Encoder." The direction of the connection in the "Updating parameter θ" diagram should be modified accordingly.

d. There are writing errors present in the article, such as all subsections after section 4.3 being numbered as 4.3. In section 4.2, Figure 5 lacks unit annotations on the horizontal axis of the CDF plot, and there is a repetition of labels in Figure 5 (a) where the CNN model is incorrectly labeled. It is essential to carefully review and correct these significant errors in the article.

There are grammatical errors in some parts of the article. Please make appropriate corrections.

Round 2

Reviewer 1 Report

Thank you for taking the time to address my comments thoroughly and comprehensively. I believe all my comments have been adequately addressed, and the quality of the manuscript has increased significantly as a result. I have determined that the manuscript is now ready for publication.

There are no major issues related to the manuscript's quality of English, except for some minor issues that do not affect the clarity and flow of the manuscript.

Author Response

We have proofread thoroughly, corrected minor errors, and improved English representation as much as possible.